# Carbon-Based Nanomaterials for Catalytic Wastewater Treatment: A Review

**DOI:** 10.3390/molecules28041805

**Published:** 2023-02-14

**Authors:** Lagnamayee Mohapatra, Dabin Cheon, Seung Hwa Yoo

**Affiliations:** 1Department of Quantum System Engineering, Jeonbuk National University, Jeonju-si 54896, Republic of Korea; 2Department of Applied Plasma & Quantum Beam Engineering, Jeonbuk National University, Jeonju-si 54896, Republic of Korea

**Keywords:** carbon-based nanomaterial, wastewater treatment, fenton reaction, photocatalysis, organic pollutants

## Abstract

Carbon-based nanomaterials (CBM) have shown great potential for various environmental applications because of their physical and chemical properties. The unique hybridization properties of CBMs allow for the tailored manipulation of their structures and morphologies. However, owing to poor solar light absorption, and the rapid recombination of photogenerated electron-hole pairs, pristine carbon materials typically have unsatisfactory photocatalytic performances and practical applications. The main challenge in this field is the design of economical, environmentally friendly, and effective photocatalysts. Combining carbonaceous materials with carbonaceous semiconductors of different structures results in unique properties in carbon-based catalysts, which offers a promising approach to achieving efficient application. Here, we review the contribution of CBMs with different dimensions, to the catalytic removal of organic pollutants from wastewater by catalyzing the Fenton reaction and photocatalytic processes. This review, therefore, aims to provide an appropriate direction for empowering improvements in ongoing research work, which will boost future applications and contribute to overcoming the existing limitations in this field.

## 1. Introduction

The demand for commercial dyes and their products has increased dramatically over the past few years. The textile, paper, and leather industries use nearly a thousand natural and synthetic organic dyes to color their products, and discharge the waste into water sources [1]. Waste discharged to water sources has been shown to contain many types of hazardous materials including, pharmaceuticals [2,3,4], dioxins [5], pesticides [6], herbicides [7], phenols [8], microorganisms [9,10,11,12], and textile dyes [13,14,15]. Therefore, water pollution has become a global problem that severely restricts the sustainable development of human society. Traditional wastewater treatment technologies have been developed, including ion exchange, reverse osmosis, oxidation, adsorption, flocculation, ultrafiltration, sedimentation, membranes, and advanced oxidation processes (AOP) [16]. Among the traditional methods mentioned above, AOP, such as the Fenton reaction and photocatalytic processes, have received extensive attention owing to their advantages of high removal efficiency, low operating cost, simple design, operation at room temperature and pressure, and environmental friendliness. 

In the Fenton process, H_2_O_2_ breaks easily, and only a natural energy source is required for photocatalysis. Fenton’s oxidation is a common, robust, and convenient AOP that is widely used for the decomposition of organic pollutants. It can be used either to lower the toxicity of wastewater or to degrade direct pollutants and allow them to enter the drainage system. Similarly, in the photocatalytic process, solid materials, called photocatalysts, accelerate the reaction rate, to eliminate the targeted pollutants from water, with the help of a UV–visible light source. To be effective, photocatalysts should have good photon-capturing properties, and the photogenerated charge carriers must have a slow recombination rate. These are the main requirements for redox reactions to complete the degradation of organic model pollutants (dyes). The requirement of a possible Fenton catalyst for the removal of aqueous pollutants in large-scale industrial applications has led to the consideration of CBMs, as they are cost-effective, eco-friendly, and sustainable. 

The unique properties of CBMs make them materials in high demand [17]. Depending on the arrangement of the carbon atoms, carbon-based materials have different morphologies. In particular, CBMs have significantly contributed toward photocatalysis because of their high surface area, outstanding conductivity, excellent chemical stability, and remarkable mechanical strength, as well as their environmental friendliness and widespread availability [18]. Considering the importance of carbon materials, their precursors must be inexpensive and effective, as well as natural and recyclable. Bio-based raw materials can be considered for all the functions listed above because of the vast variety of possible compositions and porosity scales involved [19]. Carbon exists in a wide variety of allotropic forms, from 0D to 3D nanostructures, some of which, such as graphene and its derivatives, are becoming increasingly prominent, especially when new properties are discovered and used to create specific functional CBMs for environmental applications [20]. Despite the poor catalytic activity of bare carbon in Fenton and photocatalytic reactions, CBMs with different morphologies exhibit catalytic effects. In contrast, pristine CBMs experience rapid electron-hole pair recombination and limited visible-light adsorption. Therefore, one of the best methods for resolving this issue is to construct a heterojunction, by assembling different CBMs with different dimensions [21].

Despite the poor catalytic activity of bare carbon in Fenton and photocatalytic reactions, CBMs with different morphologies promote catalytic effects. To date, various carbon-based photocatalysts have attracted tremendous interest, because the carbon atoms assemble into different dimensions and structures [22]. Therefore, carbon and carbon-based materials are extensively utilized in the cleansing of water using photocatalysis. In particular, the significance of carbon-based nanostructured materials has been recognized by the most well-known awards, including the Nobel Prize in Chemistry (1996) for fullerenes [23], the Kavli Prize in Nanoscience (2008) for carbon nanotubes [24], and the Nobel Prize in Physics (2010) for graphene [25]. Furthermore, some technologies have addressed the low efficiency of targeted pollutant removal. Therefore, several studies have focused on combining nanotechnology principles with the chemical and physical surface modification of CBMs, to produce tailored CBMs that can overcome a number of challenges faced when cleaning up pollution.

Owing to the rapid progress in the use of CBMs for the removal of organic pollutants from aqueous environments, an appropriate review is crucial to summarize the current state of the research field. Moreover, there is still no detailed review summarizing AOP-type reactions, such as Fenton and photocatalysis, using CBM for the removal of aqueous pollutants. For heterogeneous Fenton systems, different oxidants, such as H_2_O_2_, peroxymonosulfate (PMS), and peroxydisulfate (PDS), can be used to activate graphitic carbon nitride(g-C3N4) based materials for pollutant degradation in wastewater [26]. In this review article, we mainly discuss and summarize (i) several representative carbon-based (0D/1D/2D) nanomaterials; (ii) the combination of carbon-based materials with Fenton-like reactions with the particular oxidant H_2_O_2_, and photocatalytic reactions, in dye wastewater and pharmaceutical wastewater treatment; (iii) the challenges and prospects of using CBMs.

## 2. Properties of Different CBMs

Carbon is the most useful element in the periodic table. It can easily polymerize at the atomic level to form a very long carbon chain. Because there are four electrons in the outer electron shell of carbon, the carbon atoms can attach to each other by covalent single, double, or triple bonds, as well as to other elements [27]. For the above reasons, compounds composed of the same type of carbon atoms can exist in a range of molecular forms, with different properties and structures. The versatility of the arrangement of C atoms is utilized to form different allotropes, such as fullerene, graphene, and diamond [28].

Based on their size, shape, and dimension, nanostructured carbon materials are classified into different categories: 0D carbon, such as 0D fullerene, carbon dots (CDs), nondiamond (ND), and graphene quantum dots (GQDs); 1D carbon, as in carbon nanotubes (CNT) and carbon nanofiber (CNF); 2D carbon, as in graphene, graphene derivatives, graphdiynes, and graphitic carbon nitride (g-C_3_N_4_). Although all these nanomaterials contain sp2 hybridized carbon atoms, they possess different shapes depending on the hexagonal lattice arrangement. However, graphdiyne is composed of sp- and sp2-hybridized carbon atoms, whereas ND is composed of an sp3-carbon diamond core and a graphitic (sp2) carbon outer layer. Further, CDs and GQDs have both sp^2^- and sp^3^-hybridization, and g-C_3_N_4_ consists of π-conjugated graphitic planes formed via sp^2^-hybridization of carbon and nitrogen atoms (Figure 1). It is important to note that the same atomic orbitals with different shapes confer unique and distinguishable properties. In general, the physical and chemical behaviors of carbon nanomaterials are determined by their structure, dimensions, and interfacial interactions with the surrounding bulk materials. Early research focused solely on CdSe/CdS and CdSe/ZnS QD, but additional carbon-based material QDs were later created and examined. However, there was a requirement for more biocompatible QDs owing to the cytotoxicity of cadmium. C-dots mainly consist of carbon, which is an abundant and nontoxic element, and have unique structural and electrical features that distinguish them from other quantum dot families.

Owing to the specificity of the structure of carbon nanomaterials, there are several unique effects, including small-size effects, quantum effects, and surface and interfacial effects. Carbon nanomaterials can be classified into three classes based on their structure, which is well described in Figure 2. Materials that contain all the dimensions in nanoscales are 0D materials (e.g., fullerenes and carbon quantum dots); 1D materials have dimensions in the single Cartesian coordinate system, such that one of the three dimensions is >100 nm, whereas the other two are <100 nm. Two of the three dimensions are >100 nm for 2D materials; the electrons can move freely in planar motion, and only one dimension is <100 nm.

### 2.1. 0D CBMs

Among carbon-based nanomaterials, fullerenes exhibit attractive properties similar to those of CNTs and graphene for photocatalytic application. Fullerene is a 0D nanocarbon material discovered by Kroto et al., which consists of a closed-cage spherical structure with six-membered rings fused with both six-membered and five-membered rings, while five-membered rings are attached only with six-membered rings [29]. Owing to the delocalization of electrons, fullerenes are ubiquitous electron acceptors. Of the different forms of fullerenes, such as C60, C70, C76, C82, and C84, only C60 and C70 have been intensively studied. The band gap energy (Eg) of fullerene’s material is around 1.5 to 1.9 eV between the highest occupied molecular orbital (HOMO) and lowest unoccupied molecular orbital (LUMO) [30,31]. 

In addition, ND are sp2-type nanocarbons, which contain a large number of functional groups, including carboxylic acids, esters, ethers, lactones, and amines. The surface functional groups play an important role in their photocatalytic applications [32,33]. Nevertheless, the catalytic applications of ND have been less studied in the field of photocatalysis.

NDs are made up of a chemically active surface that can accommodate various functionalized moieties, and an inert insulating sp^3^ core (Eg = 5.5 eV). Historically, the mechanical reinforcement qualities of NDs have been the focus of research in this field. Through the addition of appropriate surface functional groups, their low toxicity and nitrogen/vacancy color centers have promising biomedical applications. Additionally, NDs have a large specific surface area and high carrier mobility, making it easier for photogenerated carriers to move to the photocatalyst’s surface. Therefore, NDs have been coupled with other semiconductors and widely used as photocatalysts [34]. Small carbon nanoparticles, called “carbon dots”, have diameters less than 10 nm. The properties of the carbon dots completely depend on their compositions and structures. The main distinction between carbon and quantum dots is that the former are tiny carbon nanoparticles, whereas the latter are tiny semiconductor particles with optical and electronic properties that are unlike those of the large particles. The CDs are mainly classified into graphene quantum dots (GQDs), carbon quantum dots (CQDs), and carbonized polymer dots (CPDs), according to their different formation mechanisms. CQDs are monodisperse, quasi-spherical nanoparticles composed of sp^2^/sp^3^ carbons, oxygen, hydrogen, and nitrogen, with diameters less than 10 nm. Moreover, the CQDs possess two absorption bands corresponding to π–π* and n–π* transitions for C=C and C=O bonds, respectively [35]. The fluorescence properties of CQD result from bandgap transitions of conjugated π bonds or surface defects. CQDs are chemically stable and easily surface-functionalized. They display lower toxicity, high hydrophilicity, photoluminescence emission, and high stability compared to other heavy-metal semiconductors, making them suitable for different catalytic applications [36,37]. 

Since the discovery of graphene, various graphene derivatives and graphdiyne have been extensively studied and exploited [38,39]. However, owing to its zero bandgap and low absorptivity, graphene has many limitations in practical applications. GQDs were first discovered in 2008, based on a previous study on CDs by Xu et al. [40]. Compared with other CBMs such as CD, CNTs, fullerenes, and graphene, GQDs exhibit different chemical and physical properties because of their special edge and quantum confinement effects [39].

### 2.2. 1D CBMs

CNTs are 1D allotropes of carbon. Since their discovery in 1991, CNTs have been a hot topic in carbon-based nanostructured material research and have received widespread attention in various applications, arising from them having sp2 carbon atoms with a small amount of sp^3^ carbon atoms in the form of defects [40]. CNTs are classified into single-walled CNTs (SWNTs) and multi-walled CNTs (MWNTs). In both cases, the nanotubes naturally align like “ropes” and are held together by van der Waals forces, with π–π stacking. They seem like tubular structures, and have a very high length-to-diameter ratio, typically a few nanometers in diameter and a few micrometers in length. SWNTs are fabricated by smoothly rolling single graphene sheets with a diameter of 1–2 nm. Similarly, MWNT are composed of multiple coaxial graphene layers spaced 0.34 nm apart, with diameters in the range of hundreds of nanometers. Unlike SWNTs, MWNTs usually have zero-gap energy, and their specific surface area is smaller than that of SWNTs. CNTs have attractive thermal and electronic properties that make them viable for water treatment. CNTs can act as semiconducting or metallic materials, which depends entirely on their surface structures, and exhibit excellent electronic conductivity.

### 2.3. 2D CBMs

Graphene is an example of a 2D carbon-based nanomaterial, which has a sheet-like structure and is composed of a thin layer of sp2 hybridized carbon atoms with a hexagonal shape forming a honeycomb crystal matrix. This material has been used in a variety of interesting applications. Among the many applications of 2D graphene, photocatalysis is a promising approach for fabricating different graphene-supported nanostructured catalysts for heterogeneous catalysis. Heterogeneous photocatalysis is an environmentally friendly method that has received increasing attention from the research community by using graphene derived supports [41]. Moreover, graphdiynes are a series of new 2D carbon allotropes, made up of carbon atoms with sp- and sp2-hybrids [42]. 

The structure contains acetylenic linkages (sp components) and was named graphdiyne. Since the discovery of graphene, and the prediction of graphyne, additional two-dimensional materials have received significant research attention. Among these is graphdiyne, a variant of graphyne that contains two acetylenic linkages in each unit cell rather than one, as in graphyne. Graphynes are also commonly called α-graphyne, β-graphyne, γ-graphyne, and 6,6,12-graphyne, using Greek letters for convenience, and the structural properties of sp- and sp2-hybridized carbon atoms in graphynes are determined by their structural characteristics [43]. In contrast to other carbon allotropes (CNT and graphene), which contain only sp2-hybridized carbon atoms, graphynes have sp-hybridized carbon atoms, giving them unique structures and catalytic properties [44,45]. 

g-C_3_N_4_ is a polymeric based material that consists mainly of C and N and is systematically linked by well-ordered tris-triazine-based patterns. g-C_3_N_4_ was first discovered in the year of 1834 by Liebig, who explained the preparation of linear polymers of interconnected tri-s-triazine through secondary nitrogen, naming it as “melon” [46]. After that, seven structures of C_3_N_4_ such as α-C_3_N_4_, β-C_3_N_4_, cubic-C_3_N_4_, pseudocubic-C_3_N_4_, heptazine-based structures (h-g-C_3_N_4_), orthorhombic structures (g-o-C_3_N_4_), and graphitic-C_3_N_4_ (g-C_3_N_4_) have been reported by different groups [47]. Moreover, the various electronic environments of N atoms contribute to various energy stabilities. Tris-triazine-based g-C_3_N_4_ is energetically favored and the most stable phase of C_3_N_4_ under ambient conditions, which makes it attractive for a variety of applications. The presence of sp2 hybridized carbon and nitrogen create a π-conjugated electronic structure, and are considered as a promising metal-free semiconductor photocatalyst, which can be utilized in different photocatalytic applications [48].

## 3. CBM as Fenton-like Catalysts for Wastewater Treatment 

Advanced oxidation processes (AOP) are considered to be superior to other methods for the effective removal of organic pollutants from wastewater. Fenton-like reagents represent a strong oxidizing system composed of transition metal ions such as Fe^3+^, Mn^2+^, and Ag^+^, and oxidants such as hydrogen peroxide, potassium persulfate, and sodium persulfate. CBMs have important advantages such as high flexibilities, high electrical and thermal conductivities, high mechanical strengths, and very large specific surface areas, which can make them excellent carriers for dispersing the catalyst and avoiding its agglomeration.

Carbon is the most widely used adsorbent for removing organic compounds from wastewater. On the other hand, H_2_O_2_ is a very promising disinfectant that, when combined with a catalyst, provides hydroxyl radicals (·OH) by the reduction of H_2_O_2_, thereby effecting the degradation and mineralization of pollutants. Thus, combining the adsorption capacity of carbon-based materials with the implemented catalytic activity toward the Fenton reaction is an attractive approach for wastewater treatment. In the Fenton reaction, OH^•^ can effectively attack the organic molecules to produce CO_2_ and H_2_O [49]. Fenton-like reagents can be classified into three categories as follows:
(i)Traditional Fenton reagents: The conventional system consists of H_2_O_2_ and Fe^2+^, which react with organic molecules through ^•^OH produced by the catalytic decomposition of H_2_O_2_ (Equations (1) and (2)). As the ^•^OH has a high oxidation potential, it is quickly oxidized, leading to a very fast reaction. However, many researchers have confirmed that the consumption rate of H_2_O_2_ is low, making it relatively difficult to apply directly to drinking water treatment.
Fe^2+^ + H_2_O_2_ → Fe^3+^ + ·OH + OH−(1)
Fe^3+^ + H_2_O_2_ → Fe^2+^ + HO_2_· + H+(2)(ii)Fenton-like reactions: Many researchers have worked to improve the traditional Fenton oxidation method and a large number of improved Fenton reagents such as H_2_O_2_/Fe^3+^, H_2_O_2_/O_3_, light-Fenton reagents, and electro-Fenton reagents have appeared. This method is similar to the Fenton reaction, although Fenton-like reagents were used [39].(iii)Light-Fenton reaction: This reaction introduces light sources (UV/visible light) for the Fenton reaction. The UV and Fenton catalysts can have a synergistic effect on the catalytic decomposition of H_2_O_2_, which greatly improves the oxidation of Fenton reagents. However, the amount of UV light incident at the surface of the earth is relatively low (approximately 4%), and visible light is 43%; thus, the Fenton system in the presence of light is of great significance and enhances the degree of mineralization. In this case, the Fenton-like catalyst first absorbs photons from the light source and oxidizes water molecules to produce hydroxyl radicals. Again, the electrons of the iron atoms experience charge transfer with oxygen atoms, and Fe(II) is oxidized to Fe(III) by the dissolved oxygen (O_2_) in the solution, so that HOO· and ·OH are generated during the reactions [49].(iv)Electro-Fenton method: This method is an electrochemical advanced oxidation process that can produce H_2_O_2_ when it reacts with Fe^2+^, produced by the oxidation of the Fe anode, generating ·OH and Fe^3+^. This makes use of the strong oxidizing power of ·OH to catalyze the degradation of organic matter.(v)Ultrasound-Fenton method: The ultrasound-Fenton method can be used for the pyrolysis of pollutants owing to the local high temperature and pressure generated by the ultrasonic treatment. Moreover, the strong oxidation potential of the hydroxyl radicals generated in high-temperature and high-pressure environments has an oxidizing effect on pollutants.(vi)Microwave-Fenton method: This method is similar to the ultrasound method, where the only difference is the use of microwave radiation instead of ultrasound, which promotes the decomposition of H_2_O_2_ to produce ·OH and helps to destroy the organic pollutants in wastewater.

### 3.1. Role of CBM in the Fenton Reaction

#### 3.1.1. 0D CBM 

Fullerenes are unique 0D nanocarbons with pentagonal and hexagonal rings, denoted by the formula C_20+n_, where n is an integer. As is well known, the photo-Fenton reaction arises from the photolysis of Fe^3+^ in acidic environments to produce Fe^2+^, and the subsequent interaction between Fe^2+^ and H_2_O_2_ to create ^•^OH. Zou et al. synthesized iron oxide-doped fullerene to form a C60-Fe_2_O_3_ composite for degrading RhB, MO, and phenol in the presence of H_2_O_2_ under visible-light irradiation [50,51]. This work indicated that the synergetic effects of C60 and Fe_2_O_3_ could be highly effective in heterogeneous photo-Fenton systems to enhance the photo-Fenton degradation of organic pollutants. Xu and coworkers reported using PHF/hydrous ferrite (PHF/Fh) composites for photo-Fenton activity in the simulated sunlight irradiation degradation of acidic red 18 [52]. Moreover, compared with Fh, PHF/Fh could enhance the stability and activity, because the presence of PHF/Fh could enhance the production of singlet oxygen (O_2_^−•^) and the radical OH^•^. In this context, the presence of PHF sensitizes the ground-state O_2_ to generate ^1^O_2_ and transfers electrons to Fe^3+^ on Fh, which facilitates the reduction of Fe^3+^ to Fe^2+^.

A novel family of 0D carbon materials, known as graphene quantum dots (GQDs), exhibits the characteristics of both graphene and carbon dots. In recent years, GQDs have been introduced into metal oxides to improve the catalytic performances of the catalyst. Nekoeinia et al. designed a novel nanocomposite based on CuCo_2_O_4_ and N-doped graphene quantum dots, regarded as heterogeneous Fenton-like catalysts, for MB degradation. These findings imply that GQDs significantly affect catalytic performance, and the improved activity can be attributed to the greater surface area of the CuCo_2_O_4_/N-GQD nanocomposite compared to that of pure CuCo_2_O_4_, and the synergy of N-GQDs and CuCo_2_O_4_ [53].

Moreover, CQD-based composite semiconductors are being progressively designed to be used as catalysts owing to their outstanding upconversion photoluminescence, low toxicity, and strong biocompatibility. Among them, Fe_3_O_4_@Cu_2_O/CQDs/nitrogen-doped CQDs (N-CQDs) (FCCN) were prepared by Zhang et al., demonstrating good catalytic performance and the ability to be reused, to degrade azo dyes in neutral and alkaline solutions under radiation. In this case, double quantum dots, such as CQDs and nitrogen-doped carbon quantum dots (N-CQDs), grow on the surface of the Fe_3_O_4_@Cu_2_O nanoparticles. During the photo-Fenton process, CQDs and N-CQDs, which serve as upconverting photosensitizers, increased light utilization, and the contribution of N-CQDs was greater than that of CQDs. Additionally, the CQDs and N-CQDs may transfer photogenerated charges between Cu_2_O and Fe_3_O_4_ to break H_2_O_2_ and form OH^•^ in an alkaline environment [54]. 

##### D CBM for Fenton Reaction

CNTs bearing negatively charged surface groups (e.g., carboxyl, hydroxyl, and ether) have shown great potential for enhancing the heterogeneous Fenton reactivity by combination with iron ions [55]. Several studies have confirmed that CNTs can be regarded as electron-transfer (ET) catalysts, involving the reduced and oxidized catalyst states, which affect catalytic activity. Theoretically, it was proven that, in addition to the direct injection of external electrons into the catalysts, they could also promote electron transfer between the catalyst and H_2_O_2_ and affect the redox cycle of Fe^3+^/Fe^2+^. Correspondingly, Seo et al. claimed that the enhanced decomposition of H_2_O_2_ to HO^•^ resulted from the reductive generation of Fe (II) by CNTs [56], while Gao et al. found that in the absence of H_2_O_2_, the generation of Fe(II) from Fe(III) did not change with increasing MWCNTs-COOH content, which indicates that H_2_O_2_ is essential for Fenton catalysis [57]. 

In 2017, Seung et al. fabricated magnetite iron oxide nanoparticles supported on a porous carbon nanofiber (PCNF)-based heterogeneous Fenton catalyst, by electrospinning a precursor solution, followed by heat treatment. The Fenton catalyst was composed of different phases of Fe, such as Fe_3_O_4_, α-Fe, Fe_3_C, and a trace amount of α-Fe_2_O_3_, which were dispersed on the CNF surface. In this case, CNF appeared as a cylindrical shape and the purely graphitic structure contained many defective graphitic layers, as confirmed by scanning electron microscope (SEM) and transmission electron microscopy (TEM) (Figure 3a–d). The benefits of Fe_3_O_4_ are, efficient separation from the aqueous phases, and the system follows the Haber–Weiss mechanism (Figure 3e) [58]. It has also been shown to efficiently remove methylene blue from water at various H_2_O_2_ concentrations. In this regard, Zhu et al. reported a CNTs/ferrihydrite heterogeneous Fenton catalyst for the degradation of bisphenol A and discussed the above using dynamic and thermodynamic aspects [59]. Following this work, oxidized multi-walled carbon nanotubes were combined with ferrihydrite (Fh) to form a highly efficient heterogeneous Fenton catalyst (CNTs/Fh). The results indicate that the concentrations of Fe(II), decomposition rate of H_2_O_2_, production rate of HO^•^, and catalytic reactivity of the catalysts in the heterogeneous Fenton reaction process were measured.

##### D CBM for Fenton Reaction

Graphite possesses oxygen-containing functional groups on its surface, which chemically bond with various active materials to form highly stable 2D graphene-based Fenton systems [60]. Graphene-based materials can effectively decompose hydrogen peroxide and generate hydroxyl radicals. This process has been explored extensively and has been found for many kinds of iron oxide catalysts, such as a-Fe_2_O_3_, Fe_3_O_4_, a-FeOOH, Fe_2_O_3_/carbon, and Fe0/Fe_3_O_4_ as a heterogeneous catalyst anchored with graphene-based derivatives to activate H_2_O_2_ to produce OH^•^, which can help to decompose organic pollutants [61,62]. Zero-valent iron (ZVI) is a cutting-edge material that is used for the remediation of polluted water. However, it is unstable in water and can oxidize to form a core-shell structure in which the core is zero-valent iron, and the shell is composed of iron oxides or hydroxides. Therefore, to improve its stability, a graphene sheet was wrapped with ZVI and anchored by magnetite to form an Fe^0^–Fe_3_O_4_–reduced graphene oxide (rGO) hybrid to degrade an organic pollutant (methylene blue) with a removal efficiency of 98% in 60 min (Figure 4) [63]. Then, Wang et al. fabricated Fe_3_O_4_–Mn_3_O_4_/rGO as a Fenton-like catalyst for use in the degradation of sulfamethazine (SMT) in aqueous solution [64,65]. In this case, many mechanisms were proposed that could have initiated the reaction, such as: (i) the Haber–Weiss mechanism in the metal oxide/H_2_O_2_ system; (ii) through the Fe(III)/H_2_O_2_ system; (iii) with high-valent iron species (Fe^IV^) in the metal oxide/H_2_O_2_ system; (iv) the decomposition of H_2_O_2_ on the surface of the Fe_3_O_4_, which is mostly through a non-radical mechanism. Correspondingly, the Ce0/Fe0-reduced graphene oxide (Ce0/Fe0-RGO) nanocomposite worked as a Fenton catalyst, synthesized through a chemical reduction method for the removal of sulfamethazine [64]. Many researchers have shown that Ce0/Fe0 nanoparticles tend to agglomerate to form bulky particles because of anisotropic dipolar interactions, which reduce their catalytic activities [63]. Thus, Wang et al. developed Ce0/Fe0-rGO) composites where Ce0/Fe0 was anchored on reduced graphene oxide to enhance its properties [66].

Similarly, graphidine consists of sp/sp^2^ carbon–carbon bonds and a π-conjugated structure, which is as an ideal platform for the Fenton reaction in the presence of light. Zhao et al. successfully fabricated monodisperse Fe into a porous structure of graphidine, which makes a composite with MIL-100(Fe) to form the Fe-GDY/MIL-100(Fe) photo Fenton catalyst for Dinotefuran degradation, with a maximal degradation rate of 0.055 min^−1^, which was 6.4 times higher than that of the pure MIL-100(Fe) within 60 mins. Furthermore, the mineralization rate of dinotefuran was 97.3% after 5 h, which is higher than that of other cutting-edge catalysts under the same circumstances [67]. 

g-C_3_N_4_ is a metal-free conjugated polymer 2D carbon-based nanomaterial that contains six nitrogen lone-pair electrons in its structure, which makes g-C_3_N_4_ a suitable carrier for the immobilization and dispersion of metal species, and means it has received significant attention from researchers worldwide [68]. The unique structure of g-C_3_N_4_, which contains oxygenated functional groups, helps to reduce H_2_O_2_ to form highly reactive HO^•^ radicals. The six nitrogen lone pairs of electrons that are available in g-C_3_N_4_ act as electron donors, and the adjacent carbon atom acts as an electron acceptor. Unfortunately, the Fenton-like reaction of pristine g-C_3_N_4_ for wastewater treatment is limited because of its weak redox ability, poor electron transport capacity, and low surface area. Therefore, doping with metals improves the chemical properties of g-C_3_N_4_ and promotes its further application. Moreover, in the presence of enough oxygen, g-C_3_N_4_ can produce high amounts of H_2_O_2_ by itself, which is known as the self-Fenton system and is used to degrade refractory pollutants [56,57]. In particular, heterogeneous Fenton processes have been developed by doping with transition metal ions or atoms that prolong the Fe(III)/Fe(II) cycle [58]. Recently, researchers have investigated the application of g-C_3_N_4_-based materials in in-situ Fenton reactions for degrading persistent pollutants [69,70]. Again, g-C_3_N_4_ was hybridized with an iron oxide composite to form a g-C_3_N_4_/iron oxide composite and used as a catalyst for the dark Fenton oxidative degradation of ciprofloxacin (CIP). In this case, the g-CN/iron oxide composite accelerated the Fe^3+^/Fe^2+^ redox cycle during the Fenton reaction and facilitated ciprofloxacin degradation [61]. It has been shown that metals complexed with O or N atoms of g-CN and high-density Fe–Nx centers, can change the electronic properties of CN, which shows great potential in the field of water remediation [71,72] 

Fenton-like reagents are composed of oxidants and transition metal ions (Fe^3+^, Cu^2+^, Mn^2+^, Ni^2+^, etc.), iron-containing minerals, and metal-filled polymers. Copper-based Fenton catalysts have attracted much attention because the redox performance of Cu metal is higher than Fe, which can activate H_2_O_2_ more quickly than Fe species to produce OH^•^ [73,74]. Therefore, copper oxides (CuOx) have been loaded onto defect-containing g-C_3_N_4_, which acts as a dual-reaction-center for Fenton reactions. In this case, the Cu of CuO was complexed with g-CN, which formed a dual reaction center based on the metal-O bridge [75]. Correspondingly, in situ Co-based Fenton-like catalysts were synthesized, in which Co species were modified into the framework of the graphitic carbon nitride (g-C_3_N_4_) substrate through C-O-Co chemical bonding [66]. The Fenton oxidation reaction is simple to operate and can effectively oxidize and degrade refractory organics in wastewater, but there are problems, such as insufficient mineralization of the organics, secondary pollution caused by a large amount of iron sludge, and a low utilization of H_2_O_2_ [26]. 

## 4. Carbon-Based Photocatalysts

In the case of photocatalysis, the photon energy is higher than the bandgap energy absorbed by the surface of the semiconductor, and the excitation of electrons starts, by which the electrons move toward the conduction band (CB), leaving a hole behind, subsequently forming electron-hole pairs for appropriate photocatalytic reactions [76]. In particular, CBMs are stable, highly conductive, and can be easily modified to form composites with other semiconductors to boost photocatalytic activity.

### 4.1. OD CBM Photocatalysts

The new kind of carbon allotropes such as CQDs, CDs, and GQDs are quasi-spherical, monodisperse carbon nanoparticles, with diameters below 10 nm. CQDs directly absorb light in the UV–vis range as the electrons move from the HOMO to the LUMO because of the sp2 C=C bond. As CQDs contain a variety of functional groups, most of their absorption occurs in the visible range and is a consequence of n − π* transitions. CQDs differ from other quantum dots because of the quantum confinement effect, and their bandgaps depend on their sizes. Owing to their unique structural properties, CQDs exhibit exceptional sunlight-harvesting capacities, tunable photoluminescence properties, up-converted photoluminescence (UCPL), and effective photoexcited electron transfer properties, that enable their application in photocatalysis [77]. To further adjust the PL of the CQDs, several synthesis techniques can be used to change the surface functional groups. Moreover, the photoexcitation of carbon dots is caused by π-plasmon absorption in the core carbon nanoparticles. Depending on the synthesis process, CDs can be classified into three main types: CQDs, GQDs, and carbonized polymer dots (CPDs). Under visible-light irradiation, the neat CQDs degraded methylene blue in aqueous solutions [68]. Although few reports have been published on CQD photocatalysts alone, they can increase the photocatalytic activity of other photocatalysts by acting as electron mediators, photosensitizers, and/or spectrum converters [78].

The advantages of using natural resources for the synthesis of CQDs are that they are inexpensive and environmentally friendly. The development of novel strategies using natural carbon sources is encouraged. Moreover, heteroatom-doped (S- and N-doped) graphene quantum dots containing sulfur and nitrogen (S- and N-doped GQDs) were physically combined with titanium dioxide to create S- and N-QGD/TiO_2_ composites for Rhodamine B photocatalytic degradation. This combination showed ten times better results than the existing P25 TiO_2_ [79]. Similarly, ZnO/S and N-GQDs exhibit high dye degradation efficiencies [80]. There are several functions that CQDs play in effective photocatalysis: (i) serving as an excellent mediator and acceptor for electrons produced in the conduction band (CB), (ii) improving the broad visible absorption band of the photocatalyst’s visible light activity through photosensitization techniques, (iii) acting as a reducing agent during the formation of metal nanoparticle surface plasmon resonance (SPR) phenomena of different metal nanoparticle production, and (iv) effectively harvesting a wide solar spectrum using up-conversion photoluminescence (UCPL) phenomena. Therefore, the bandgap, valence band, and conduction band depend completely on each other for the excitation and photoactivity of photocatalysts modified with CQDs [81]. Since GQDs have a high degree of crystallinity, they have better electron transport properties and longer carrier lifetimes. In addition, C60 are thought to be excellent electron acceptors and transporters, with a narrow band gap (roughly 1.6–1.9 eV). They can directly be used as energy transfer mediators to produce _1_O^2^ for reaction. Because of their unique structure, they have great applications in many fields, such as photovoltaics and photocatalysis [82,83].

### 4.2. 1D CBM Photocatalysts

CNTs and CNFs are ideal ID support nanomaterials that possess the characteristics of both nanomaterials and fibrous structures. Owing to their large surface areas, intriguing electronic properties, unique physicochemical properties, and high aspect ratios, CNTs and CNFs are promising candidates for the design and synthesis of novel photocatalysts. In general in CNT-based photocatalyst systems, CNTs act as “electron sinks” that receive photogenerated electrons from the semiconductor, thereby realizing the effective separation of photogenerated electron-hole pairs [84]. They also facilitate electron transport in heterogeneous photocatalyst systems through “electron bridges.” It is worth mentioning that some researchers observed that the absorption edge of CNT-based photocatalysts shifts to a higher wavelength than that of bare photocatalysts. CNTs can also be functionalized via surface oxidation or heteroatom doping. This significantly facilitates the potent connection and homogeneous distribution of semiconductor nanoparticles on the CNTs. Sometimes, when the tube diameter is smaller than 6 nm, the SWCNTs can be regarded as 1D quantum wires with good conductivities [85]. It should be noted that untreated CNTs contain various amorphous carbons, graphite fragments, and other impurities, which is not conducive for widespread application. In addition, it is difficult to combine them with other semiconductors to form heterostructures. In the case of CNTs, the important factor is the interfacial contact between the CNT and another semiconductor, which favors the transmission of electrons and reduces the recombination of charge carriers. Therefore, it should be functionalized by surface oxidation or metal/non-metal doping. To date, various CNT-based photocatalysts that can utilize solar energy to degrade various pollutants into non-toxic substances have been constructed [86,87,88,89,90,91]. However, compared to CNTs, CNFs may have a higher aspect ratio, which gives them far superior assimilability. In addition, by employing proper synthesis procedures, the porous structure and composition of the CNFs can be easily controlled. In particular, the electrospinning method has become a versatile technique for fabricating CNFs because of its easy operation, low cost, and feasibility for a wide range of carbon precursors. As a result, the electrospun CNFs have drawn significant research attention in the field of photocatalysis, and many studies have been reported [92,93].

### 4.3. 2D CBM Photocatalysts

#### 4.3.1. Graphene-Derived Materials for Photocatalytic Wastewater Treatment 

Graphene, graphene oxide (GO), rGO, and exfoliated GO are known as graphene derivatives that have become promising materials for wastewater treatment owing to their favorable characteristics [94]. Graphene has been thoroughly investigated as a favorable photocatalyst owing to its large surface area, high electrical conductivity, and low cost for mass production. Moreover, graphene is a zero-bandgap semiconductor, and the VB and CB of graphene consist of bonding π and anti-bonding π (π*) orbitals, which touch the Brillouin zone corners, making a single sheet of graphene a zero-bandgap semiconductor [95]. Graphene and its derivatives are used as stand-alone photocatalysts, co-catalysts in composites, or as supporting materials for semiconductor catalysts. When graphene and its derivatives are coupled with conventional photocatalyst containing metals and metal or nonmetal oxides/sulfides/phosphides, these materials are advantageous for water treatment [96]. The proper combination of 1D semiconductor materials on the surface of 2D graphene materials can also prolong electron-hole recombination, owing to their outstanding electrical conductivity. This electronic transport property supports catalytic activity by improving the transport of electrons to allow redox reactions. Among the most reported graphene-based composites, binary and tertiary composite systems, consisting of metal oxides with graphene derivatives, have shown considerable potential for the remediation of organic compounds owing to their unique layered electronic band structures, optical properties, and narrow visible-light absorbance.

#### 4.3.2. Graphdiyne Based Materials for Photocatalytic Waste Water Treatment 

Graphdiyne has attracted significant interest in photocatalysis because of its distinctive fundamental characteristics. High-performance photocatalytic degradation has been extended to graphdiyne-based catalysts [97]. Given the variety of ligands, graphdiyne exhibits structural variations and tunable characteristics. These materials showed semiconducting properties with a direct band gap of 0.46–1.22 eV, which induces the occurrence of the photocatalytic process when exposed to light energy. Moreover, their large surface areas and porous structures allow for rapid mass transfer and the exposure of more catalytic sites, accelerating the catalytic reaction process. Even under specific reaction conditions, such as acidic and alkaline environments, these materials demonstrate exceptional chemical stability during the catalytic reaction, as described by Wong et al. [98]. Therefore, graphdiyne-based materials are suitable platforms for photocatalytic reactions. To date, numerous experimental studies have been devoted to the development of GDY-based heterojunctions with other semiconductors, such as TiO_2_, for the removal of photocatalytic organic pollutants. In this case, the graphdiyne and TiO_2_ nanoparticles formed a Ti-O-C link. As reported, the impurity bands of the carbon p orbitals caused the composite to exhibit visible light photocatalytic activity for the degradation of methylene blue dye. In this process, graphdiyne served as an acceptor for the photogenerated electrons, which helped to reduce charge recombination and accelerated the degradation of the dyes [99].

#### 4.3.3. g-C_3_N_4_ Derived Materials for Photocatalytic Waste Water Treatment

In the field of heterogeneous catalysis, g-C_3_N_4_ was introduced as a photocatalyst in 2006, and was recognized for its high chemical and thermal stability, suitable bandgap energy of 2.7 eV, with a −1.1 eV of CB position and 1.6 eV of VB position vs. normal hydrogen electrode (NHE) [48]. Generally, in photocatalytic reactions, semiconductors can produce electron and hole pairs in the presence of light, and the main active species, such as H+, ^•^OH, ^•^O_2−_, are required. When dissolved oxygen reacts with electrons, ^•^O_2−_ radicals can be produced. However, ^•^OH radicals originate from two pathways: the direct oxidation of H_2_O/OH^×^ by photoexcited holes (the oxidation potential is 1.99 eV), and the disproportionation reaction of ^•^O_2−_ radicals. However, its disadvantages include its practical applications, owing to its lower visible-light absorption capacity, smaller specific surface area, lower quantum yield, and high recombination rate due to weak van der Waals interactions between adjacent carbon nitride layers. Therefore, researchers have focused on altering the coupling of g-C_3_N_4_ with other semiconductors, which changes the electronic characteristics via band assembly engineering. Moreover, increasing the crystallinity of g-C_3_N_4_ is an effective way to reduce the recombination of photogenerated charges and enhance its photocatalytic performance [100]. It has also been confirmed that the catalytic activity is extremely dependent on the morphology and size, which can extend the visible-light absorption capacity [101]. Various strategies have been developed to modify g-CN and improve its photoactivity. According to earlier research, the improved photocatalytic performance of g-C_3_N_4_ can be attributed to structural defects such as nitrogen or carbon vacancies or sp2 nitrogen defects. Some authors have used various organic nitrogen-rich organic precursors that have been utilized for the preparation of carbon nitride through thermal polymerization. Moreover, the microstructure of materials and their photoactivities are strongly related [102]. The microstructure of the catalyst and synergetic reactions of the catalyst-associated reaction pathways were clarified [103]. Therefore, the interactions between components depend on the methods and conditions used to obtain the nanocomposites.

First, the surface modification and creation of defects in g-C_3_N_4_ widened its visible-light absorption and prevented the recombination of charge carriers [102]. Usually, the defects on g-C_3_N_4_ contain C and N defects that can create reaction sites for the reactant molecules, broaden their visible-light absorption range, and prevent the recombination of charges. The defects arise from the breakage and interruption of the regular arrangement of basic units in g-C_3_N_4_, which influences the photocatalytic activity. Surface defects engineered on g-CN contain C and N defects, which broaden the visible light absorption area and prevent charge recombination. N-defective g-C_3_N_4_ with an adjustable band structure has been reported as an impressive visible light photocatalyst for water remediation [104]. The activity of g-C_3_N_4_ also depends on structural engineering, such as the shapes, dimensions, and sizes of nanosheets, nanowires, nanotubes, nanorods, and nanobelts [105]. Interestingly, 2D sheets, which are well-defined building blocks, play a vital role in photocatalysis compared to bulk g-C_3_N_4_. The second strategy is the modification of g-CN by doping with metal/nonmetal elements, which leads to a noticeable modulation of the light absorption capacity [106,107]. The third method is the formation of heterojunctions when g-C_3_N_4_ is combined with other types of semiconductor photocatalysts with suitable energy levels, which has been widely used for photocatalytic dye degradation [108,109,110,111].

Notably, the introduction of CDs into g-C_3_N_4_ significantly enhanced its catalytic performance. Owing to their small dimensions, ability to absorb visible light, and up-conversion photoluminescence properties, CDs exhibit electron transfer/reservoir properties, making them excellent materials when combined with another semiconductor for photocatalysis. Recently, many groups have prepared metal-free CDs/g-C_3_N_4_ hybrid photocatalysts for various photocatalytic applications [112,113]. Li et al. fabricated CQD-decorated g-C_3_N_4_ by a novel strategy employed for rhodamine B (RhB) degradation under visible light irradiation [114]. This study confirmed that the dye degradation rate of g-C_3_N_4_/CQDs nanocomposites (0.013 moL^−1^min^−1^) was three times greater than that of bare g-C_3_N_4_ (0.0044 moL^−1^min^−1^), as shown in Figure 5a,b. The enhanced photocatalytic activity can be ascribed to the CDs helping the efficient electron transfer from the conduction band of g-C_3_N_4_ to the adsorbed O_2_, forming O_2_^•^.

Yang et al. studied a metal-free g-C_3_N_4_/CD heterojunction synthesized by directly coating CDs onto the g-C_3_N_4_ surface for phenol degradation under visible-light irradiation, and the role of CD was systematically investigated [115]. For instance, N doped carbon quantum dots with defective g-C_3_N_4_ (NCDs/DCN) were prepared using the impregnation method. The resulting hybrid catalyst exhibited outstanding removal efficiency for different combinations of pollutants, including OFL/Cr (VI), BPA/Cr (VI), and CIP/Cr (VI). The improved photocatalytic activity of the hybrid system was due to its electron transfer ability and the extension of the visible-light absorption region [116]. Correspondingly, a sulfur-doped CQD/g-C_3_N_4_ composite was synthesized by a simple in situ polymerization method for methyl orange degradation, which showed much better photoactivity performance than g-C_3_N_4_ [117]. Moreover, abundant research on CD/g-C_3_N_4_-based composites has focused on the removal of antibiotics from water. Liu et al. reported a g-CN/CQD composite prepared using a thermal polymerization approach for diclofenac degradation under visible-light irradiation. The fundamental mechanism of the photocatalytic degradation of organics with specific structures was discussed using DFT calculations [118]. It was finally confirmed that the modification of CDs did not have an obvious effect on the morphology and structure of g- C_3_N_4_, whereas it considerably influenced the optical properties, so that the CDs/g-C_3_N_4_ composite could be applied for water remediation. Very recently, our group studied visible-light-induced N-rich g-C_3_N_4_ (NCN)/N-doped CQDs (NCD) hybrid catalysts for the simultaneous oxidation of phenolic compounds, such as 4-nitrophenol (4NP), and the reduction of Cr(VI), in the presence of visible-light irradiation. The presence of N-doped CQDs on the surface of defect-rich g-C_3_N_4_ with a lattice spacing of (002) plane was observed in a high-resolution transmission electron microscopy (HRTEM) image to form a hybrid photocatalyst (Figure 6a). In particular, the 2.0% NCD/NCN catalyst exhibited outstanding photoactivity for the coupled removal of 4NP (95%) and Cr (VI) (97%) within 60 min. Also, 2NCD/NCN possessed a smaller band gap energy (1.99 eV), with −0.34 eV and +1.65 eV, with CB and VB positioned, respectively. Moreover, the highest intensity of the ESR signal of 2NCDs/NCN suggests higher amounts of OH^•^ and O_2_^−•^ (Figure 6b). Thus, it was confirmed that the enhanced photocatalytic activity of the optimum catalyst is due to the synergistic effects of reduced e- and h+ recombination rates, improved transient photocurrent density, lower band gap energy, and superior up-conversion photoluminescence properties. Moreover, the improved electron delocalization features were confirmed by ESR experiments. Additionally, the higher photocatalytic efficiency of the mixed pollutant system was attributed to the dual functions of 4NP oxidation and Cr(VI) reduction, which are important for efficient charge separation [101]–the details are described in Figure 6c [119].

#### 4.3.4. Multidimensional Hybrid CBM-Based Photocatalysts

Carbon-based hybrid nanomaterials (0D–3D) with various dimensions have attracted considerable attention because of their unique structural, electronic, and optical properties, and easy functionalization. Moreover, the contact between multidimensional carbon-based matrices, which can form efficient heterojunctions, ensures the formation of high-quality interfaces, resulting in high-efficiency charge transfer. The different morphologies of heterostructures such as point to point contact {(0D-0D), (0D-1D)}, point-to-face contact (0D-2D), line to face contact {(1D-1D), (1D-2D)}, and face-to-face contact (2D-2D) provide an excellent support matrix (Figure 7). In particular, the efficient heterojunctions with intimate contacts between them, such as point-to-face contact (0D-2D), line-to-face contact (1D-2D), and face-to-face contact (2D-2D) are used for many catalytic applications. This is beneficial for rapid charge transfer and better catalytic dispersion to enhance photocatalytic activity. The enhanced photocatalytic activity depends upon the morphology-based heterojunctions for photocatalytic water treatment.

Zero-dimensional carbon-based materials have shown great potential when modified with 2D carbon-based materials, owing to their unique characteristics, including excellent electrical conductivity and stable physicochemical properties, which favor charge-transfer kinetics. Therefore, the synergetic interaction between C60 and g-CN in the C60/g-C_3_N_4_ based composite was attributed to the improved photoactivity toward wastewater treatment. Similarly, the 0D/2D-based CD/g-CN heterojunction is a promising photocatalyst owing to its tunable bandgap and efficient charge transfer during photocatalysis, which has already been discussed [120]. As CNTs have received considerable attention owing to their unique properties, they have been used in various composites for photocatalytic applications. Hybrid 1D with 2D semiconductors appear to be an effective way to increase the photoactivity of CN. Better improvement was achieved by designing a metal-free photocatalyst containing CNT and C_3_N_4_. The CNT/C_3_N_4_ hybrid catalyst showed a better performance toward the degradation of the MB dye solution owing to its light absorption capacity, and it retarded the recombination of charge carriers [121]. In general, functionalization is required to fabricate heterojunction catalysts, because a large number of oxygen-containing functional groups are available in GO. In addition, the combination of CN with graphene derivatives creates a 2D/2D hybrid carbon-based catalyst, which is considered a noteworthy catalyst for improving photocatalytic performance [122]. In particular, 2D–2D based CN/Graphene heterojunctions are particularly advantageous because of their high catalytic surface area and abundant reaction sites, as well as the suppression of the recombination of photogenerated electron-hole pairs at the 2D/2D interface and facilitation of charge transfer. In this context, we can conclude that (i) strong interfacial contacts; (ii) more active sites; (iii) low charge transfer resistances; (iv) shortened diffusion paths; and (v) improved light absorption capacities result from ultrathin 2D materials [123,124].

## 5. Conclusions and Prospects

The explosive growth in the field of carbon-based nanomaterials in recent decades has unquestionably revealed novel interesting properties, with remarkably improved catalytic activities. These materials exhibit extraordinary thermal conductivities, and mechanical and optical properties. CBMs are widely used as support materials for catalysis. In this review, we addressed various innovative carbon-based (0D, 1D, and 2D) catalysts with unique characteristics and advantages. We highlighted their widespread utilization in the area of catalysis, such as Fenton-like reactions and photocatalytic applications for wastewater treatment. 

In general, the development of different CBMs was accompanied by three main far-reaching factors: expanding the surface area; suppressing the recombination of photogenerated electron-hole pairs; and increasing the number of active sites on the surface of the supported materials. Based on these criteria, different dimensions of CBM (0D–3D) as a class of promising materials have been applied in the modification of hybrid materials, and they have become outstanding materials owing to their cost-efficiency, excellent electrical conductivity, appealing optical properties, etc. Moreover, the different structures of specific CBMs offers both thermodynamic and kinetic control. Although advancements in CBMs with different dimensions have been studied in recent years, some issues still need to be addressed.

Fenton-like reactions are currently in use for the treatment of aqueous organic pollutants; however, further investigation of the reaction mechanism and its efficient application is required. Moreover, novel and inexpensive catalysts that do not cause secondary pollution and that behave in an environmentally friendly manner should be developed.In addition, current research is mainly focused on the multi-component hybridization of 0D/1D/2D CBM and other CBMs for photocatalytic applications to improve photocatalytic performance. However, the development of 3D CBMs with other carbonaceous materials is still lacking. Therefore, future research can attempt to couple 3D CBMs with carbonaceous materials of various dimensions, which can be expected to bring new prospects and stronger photocatalytic performance.Compared to traditional metal-based modifiers, carbonaceous materials are relatively simple and cost-effective. However, it is worth noting that the cost of some carbon materials, calculated on a laboratory scale, such as 0D-CD are still far from being viable for large-scale applications. Therefore, the development of low-dimensional low-cost carbon materials for hybrid CBMs is not only the exploration of new synthetic CBMs but also the source of raw materials selected in the future.As photocatalytic applications involve many steps, such as photon absorption, charge formation, charge carrier separation and recombination, surface reactions for radical generation, and targeted reactions, the final photo efficiency is accurately controlled by each step and should be integrated on an experimental and theoretical basis.Despite the substantial advances in the synthesis and catalytic activity of green catalysts, future research should focus on reducing the cost of the synthesis of CBM-based materials and improving their catalytic efficiency.Biowaste-derived CBMs should be used in wastewater treatment processes.Magnetic CBMs should be considered for the synthesis of materials, in order to increase their activity and potential for reuse.

Currently, research is concentrated on CBMs and their production from waste materials and natural precursors. CBMs have demonstrated a tremendous potential for water treatment and purification, particularly for removing pharmaceutical and industrial pollutants. Overall, CBMs as Fenton catalysts or photocatalysts function as a blue ocean, and high activity, high stability, and high efficiency are the ultimate goals for wastewater treatment. This review provides a valuable reference for future research into wastewater treatment.

## Figures and Tables

**Figure 1 molecules-28-01805-f001:**
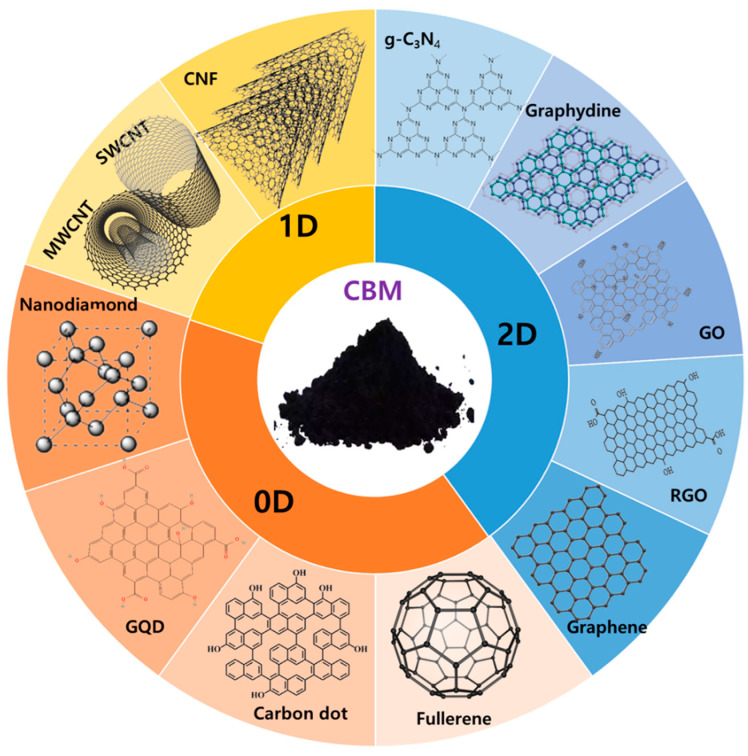
Graphical representation of the different dimensions of carbon-based nanomaterials.

**Figure 2 molecules-28-01805-f002:**
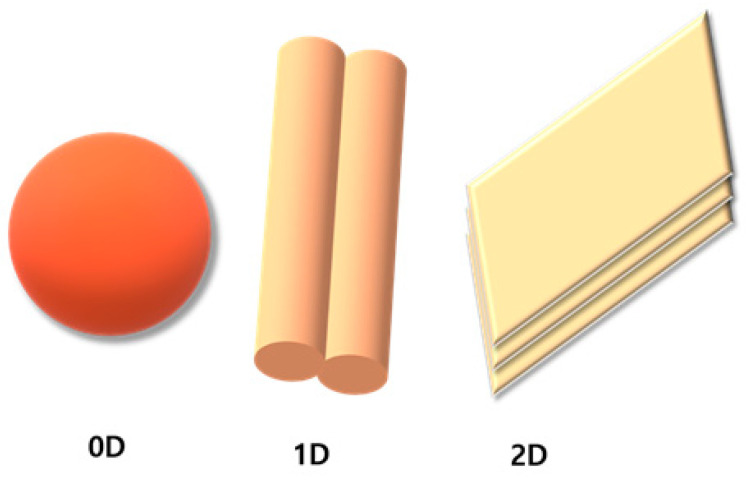
Graphical representation of various dimensional materials.

**Figure 3 molecules-28-01805-f003:**
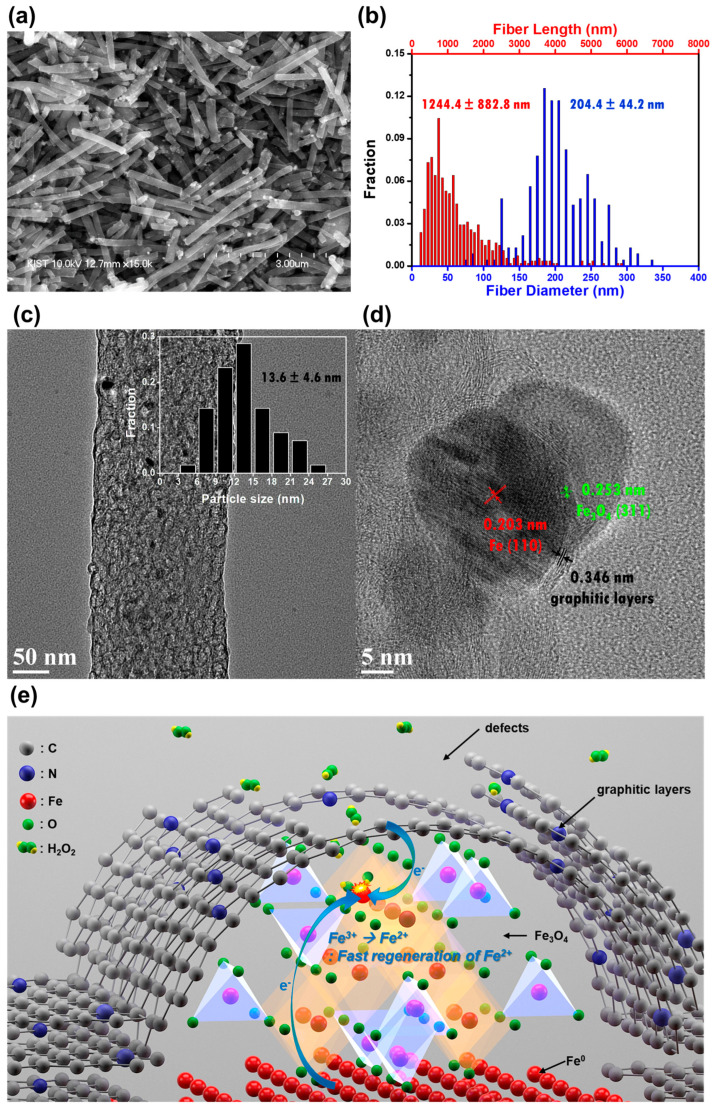
SEM image of Fe_3_O_4_/Fe/Fe_3_C@PCNF, (**b**) The distribution of length and diameter of fibers in (**a**), (**c**) Low-magnification TEM image of Fe_3_O_4_/Fe/Fe_3_C@PCNF. Dark spots correspond to the catalyst nanoparticles dispersed throughout PCNF. The inset displays the size distribution of these nanoparticles. (**d**) High-magnification TEM image of the vicinity of catalyst particles positioned at the outermost edge of PCNF. (**e**) The suggested mechanism of extremely fast H_2_O_2_ catalytic decomposition by Fe_3_O_4_/Fe/Fe_3_C@PCNF. Reproduced with permission from RSC from [58].

**Figure 4 molecules-28-01805-f004:**
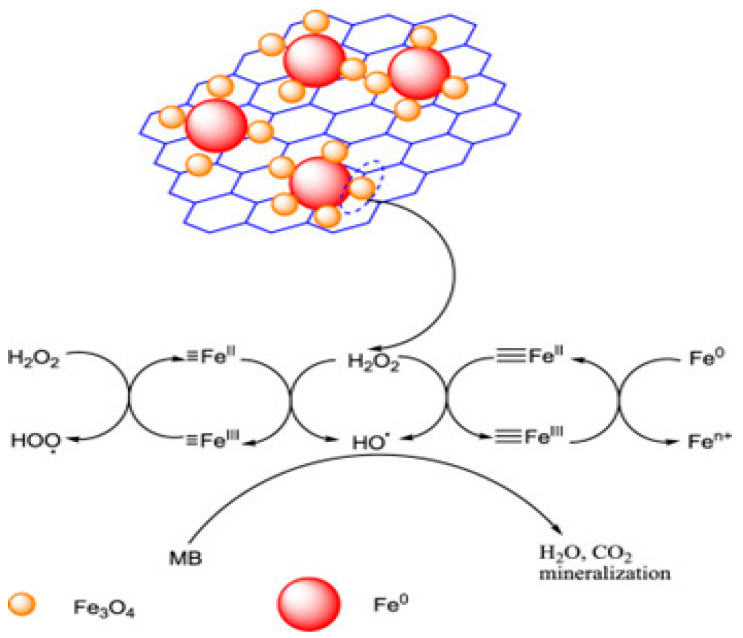
The possible Fenton catalytic oxidation process. Reproduced with permission from Elsevier [63].

**Figure 5 molecules-28-01805-f005:**
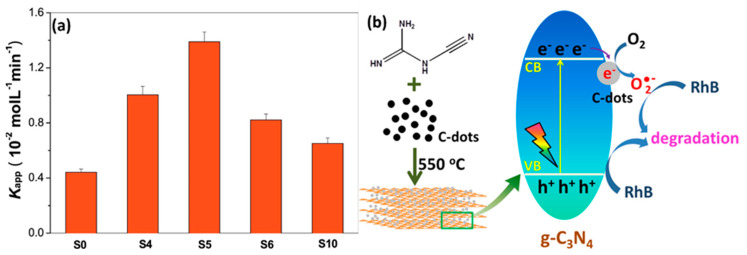
(**a**) Comparison of the degradation rate constant of C-dots-modified g-C_3_N_4_ (**b**) Schematic diagram showing the one-pot synthesis of C-dots-modified g-C_3_N_4_ with enhanced photocatalytic activity. Reproduced with permission from Elsevier [113].

**Figure 6 molecules-28-01805-f006:**
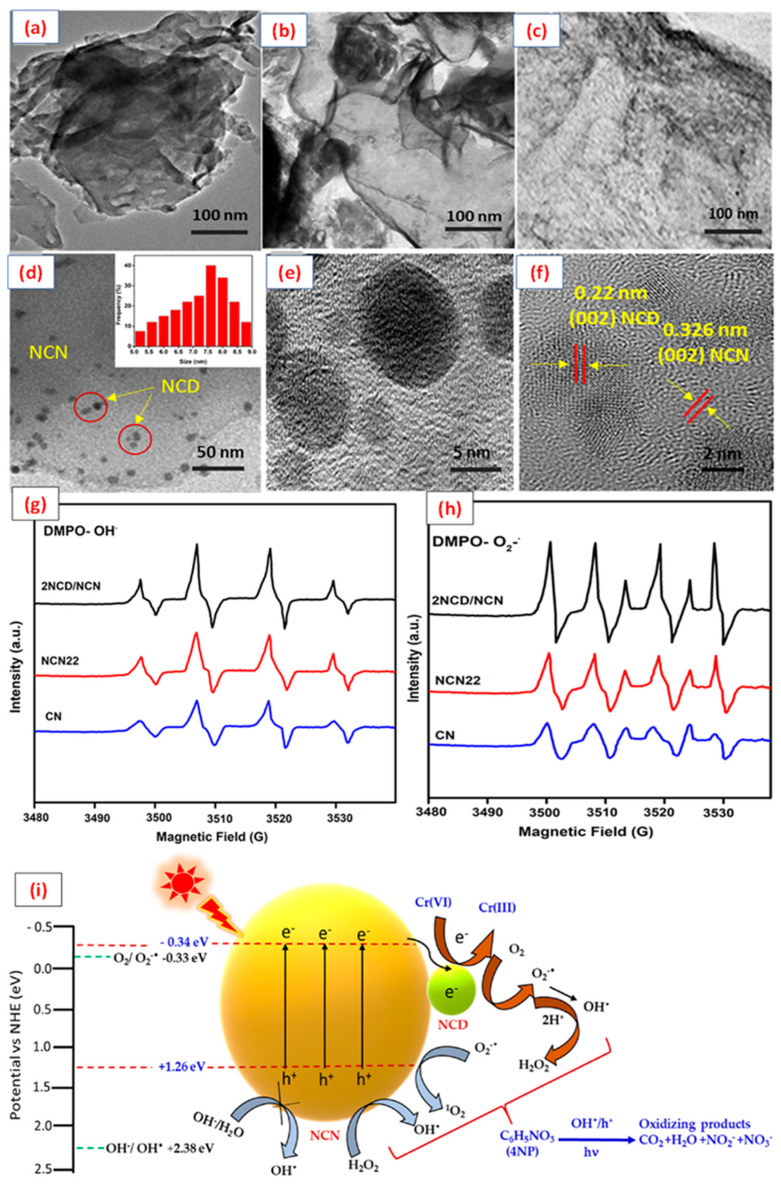
TEM (**a**–**c**) and HRTEM (**d**–**f**) images of CN, NCN22, and 2NCD/NCN. ESR spectra of radical adducts trapped by DMPO in C_3_N_4_, NCN22, and 2NCD/NCN under visible-light irradiation in (**g**) aqueous dispersion for DMPO-OH• and (**h**) methanol dispersion for DMPO-O_2_•−. (**i**) The proposed photocatalytic mechanism for the simultaneous degradation of 4NP and reduction of Cr(VI) under visible-light irradiation. Reproduced with permission from Elsevier [119].

**Figure 7 molecules-28-01805-f007:**
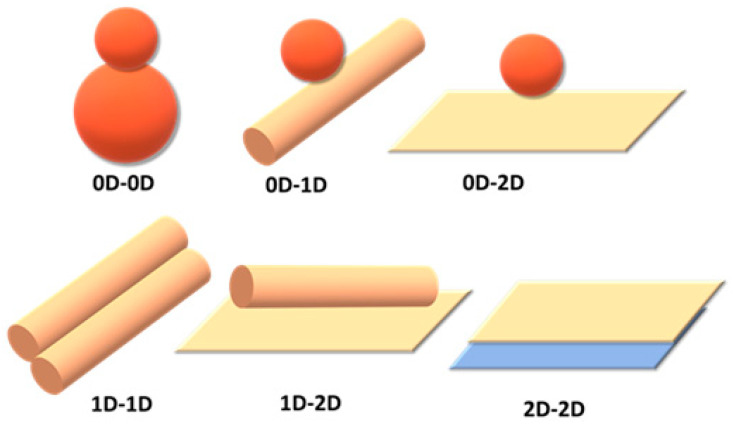
Schematic representations of various interfaces combining various dimensional mate–rials.

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
