# Peer review of "Carbon-Based Nanomaterials for Catalytic Wastewater Treatment: A Review"

_molecules, 2023, doi:10.3390/molecules28041805_

Round 1

Reviewer 1 Report

File attached

Author Response

Reviewer 1

The manuscript titled “Carbon-based Nanomaterials for Catalytic Wastewater Treatment: A Review” is an interesting work. The review described he contribution of carbon based

nanomaterials, with different dimensions, to the catalytic removal of organic pollutants from wastewater by catalyzing the Fenton reaction and photocatalysis process. The review article can be proven of certain significance. This work could be considered for publication and prove to be more interesting if the authors made the following changes/modifications.

  1. The background of catalytic wastewater treatment response of carbon-based materials needs to be further strengthened.

Answer: We thank the reviewer’s valuable comments and thoughtful suggestions regarding our manuscript. We have explained in the introduction part of the revised review manuscript.

  1. The author needs to clarify the influence of material microstructure on the performance mechanism.

Answer: We thank the reviewer for their valuable suggestions and insert the influence of material microstructure on the performance mechanism in the 4.3.3 section.

  1. The progressive relationship between each work is not clear, and the overall logic of the article is poor, which needs further combing. The innovation points of each work were not well reflected, and the reasons and mechanisms for the performance improvement were not well explained.

Answer: Thank you for these suggestions and we have added the mechanism of each work in each section.

  1. Some important work needs to be introduced in the content, e.g.

https://www.mdpi.com/2079-4991/11/10/2548); DOI: 10.1039/D2VA00018K

Answer: Thank you so much reviewer’s valuable comments. We have inserted the references (refence 14,15) in the reference section. 

  1. The conclusion and outlook are not deep enough to guide peers. The problems encountered in the current research, especially the regulatory mechanism, should be analyzed in more detail. This section should be considered carefully and highlighted.

Answer: Thank you so much. We improved the conclusion part. Also, We explained the novelty and findings of revised review manuscript in the introduction section and conclusion section.

  1. The author needs to recheck headings and sub-heading and their numbering, for example 2.1.0. D CBMs, 2.2.1. D CBMs, 2.32. D CBMs, etc.

Answer: Thank you so much. We checked carefully and corrected.

  1. There are some language errors and description errors in the article, which can be modified to make the article more beautiful.

Answer: Thank you so much reviewer’s valuable comment. We check very carefully and changed the sentences where it needs.

Reviewer 2 Report

The manuscript deals with the Carbon-based Nanomaterials for Catalytic Wastewater  Treatment: A Review. The manuscript is well structured, however some points should be clear for better understanding of the presented data for possible publication in Molecules.

A.    In introduction, the author should clearly mentioned the utiliziation of Carbon based materials and their importance. The author should carefully read the articles like Advanced Powder Technology 32 (2021) 3770–3787; Environmental Research 215 (2022) 114140 and refine this portion accordindly.

B.     In the introduction section, the author should include the name of targeted dyes, their role, and their hazardous effect on the ecosystem through carefully reading the article Advanced Powder Technology 33 (2022) 103708, https://doi.org/10.1007/s11356-022-22144-3.

C.     The author should clearly explain the enhanced efficiency of the current materials under study.

D.    All figures should be in same style and size.

E.     It is very difficult and expensive to synthesize carbon based material, what is the advantage of using these materials?

F.      Explain the mechanism of photocatalysis using these materials.

G.    What is the novelity of the present findings?

Author Response

Reviewer 2

Comments and Suggestions for Authors

In this review the contribution of carbon-based nanomaterial with different dimensions to the catalytic removal of organic pollutants from wastewater via Fenton reaction and photocatalysis process. Overall, this review is comprehensive, this review shall be attractive to the committees of environmental catalysis and nanomaterial design. Several concerns shall be addressed before publication:

  1. The category of carbon nanomaterials is not very accurate throughout this manu, the authors should pay more attention to this section, to name a few:
  2. Line 85, 125 and 361, carbon dots should include graphene quantum dots.

Answer: Thank you so much Thank you so much reviewer’s valuable comments. We checked carefully and included GQD.

  1. Line 86, 1D carbon should encompass carbon ribbon, though such nanomaterials are not widely used in organic pollutant removal. However, nanodiamond was not discussed for removal of organic pollutant in this review or listed in Figure 1, but was included in 0D carbon in line 85.

Answer: Thank you so much reviewer’s valuable suggestions. We have included the details about nanodiamond in 2.1 section, and new references added in reference section as 29,32and 33.

  1. Figure 1, there are many structural or properties overlap between graphite oxide, reduced graphene oxide, and graphene, I would suggest the authors to classify them into two types, one is graphene, another is graphene derivative. If possible graphydine shall ben be also included, if such type 2D materials were used in the removal of pollutants from wastewater.

Answer: Thank you so much for the reviewer suggestions. We have modified the figure 1 and included graphydine. Also we have added 2D materials as graphydine for the removal of pollutants from wastewater in section 2.3 and 4.3.2.

  1. Line 13, “which are not yet matched by inorganic systems.” Many conventional carbon-based nanomaterials are regarded as inorganic materials, the authors shall pay more attention to this sentence.

Answer: We appreciate the reviewer’s comments and modified the sentence with appropriate references.

  1. The subtitle is bit misleading in Line 110, 143, 159, 231, 245, 281, 355, 369, 389. The author shall just use 0D, 1D and 2D.

Answer: We appreciate the reviewer’s comments. We corrected those above parts.

  1. I noticed that the sentence in Line 142 “which are smaller than 100 nm in size and less than 10 layer thick” was also mentioned in Ref 30, however, the quantum confinement emerges generally in sub 10 nm, the author shall further confirm the accuracy of this sentence.

 Answer: Thank you so much. We corrected those part with appropriate references.

  1. The chemical formulas in this manu are very unformal, particularly for the subscript.

Answer:  Thank you so much for the suggestions. We have checked carefully the formulas and have done the needful corrections.

  1. Line 269, the sentence was wrongly correlated with Figure 2. 

 Answer: Thank you so much. We corrected.

  1. Figure 6, the panel captions are also confusing, the author shall remove the initial caption like (a), (b) and etc.

Answer: We thank the reviewer for their valuable comments and thoughtful suggestions regarding our manuscript. We have modified the captions.

  1. Line 359, the author claims that “CQDs are different from other quantum dots because of their quantum confinement effect, and the bandgap depends on their size”, such a nanosized effect is generic for many nanomaterials, the author shall interpret more of include related references.

Answer: Thank you so much. CQDs are different from other quantum dots.

  1. Line 232, the sentence “Fullerenes have a unique three-dimensional π-electron delocalization structure” is a bit misleading, the author shall further clarify more details.

Answer: Thank you so much. We clarified those parts in the 2.1 and 3.1 section of the revised manuscript.

  1. Several typos or vagueness shall be further clarified:
  2. Line 120, ND are sp2-type nanocarbons

Answer: Thank you so much. We Corrected in the revised manuscript.

  1. Line 493, a lattice spacing of 002

Answer: Thank you so much. We checked very carefully and explained in the revised manuscript.

Reviewer 3 Report

In this article, Tae-hyun Kim et al., review the contribution of carbon-based nanomaterial with different dimensions to the catalytic removal of organic pollutants from wastewater via Fenton reaction and photocatalysis process. Overall, this review is comprehensive, this review shall be interesting to the committees of environmental catalysis and nanomaterial design. Several concerns shall be addressed before publication:

1.      The category of carbon nanomaterials is not very accurate throughout this manu, the authors should pay more attention to this section, To name a few:

a.       Line 85, 125 and 361, carbon dots should include graphene quantum dots.

b.      Line 86, 1D carbon should encompass carbon ribbon, though such nanomaterials are not widely used in organic pollutant removal. However, nanodiamond was not discussed for removal of organic pollutant in this review or listed in Figure 1, but was included in 0D carbon in line 85.

c.       Figure 1, there are many structural or properties overlap between graphite oxide, reduced graphene oxide, and graphene, I would suggest the authors to classify them into two types, one is graphene, another is graphene derivative. If possible graphydine shall ben be also included, if such type 2D materials were used in the removal of pollutants from wastewater.

d.       Line 13, “which are not yet matched by inorganic systems.” Many conventional carbon-based nanomaterials are regarded as inorganic materials, the authors shall pay more attention to this sentence.

2.      The subtitle is bit misleading in Line 110, 143, 159, 231, 245, 281, 355, 369, 389. The author shall just use 0D, 1D and 2D.

3.      I noticed that the sentence in Line 142 “which are smaller than 100 nm in size and less than 10 layer thick” was also mentioned in Ref 30, however, the quantum confinement emerges generally in sub 10 nm, the author shall further confirm the accuracy of this sentence.

4.      The chemical formulas in this manu are very unformal, particularly for the subscript.

5.      Line 269, the sentence was wrongly correlated with Figure 2. 

6.      Figure 6, the panel captions are also confusing, the author shall remove the initial caption like (a), (b) and etc.

7.      Line 359, the author claim that “CQDs are different from other quantum dots because of their quantum confinement effect, and the bandgap depends on their size”, such a nanosized effect is generic for many nanomaterial, the author shall interpret more of include related references.

8.      Line 232, the sentence “Fullerenes have a unique three-dimensional π-electron delocalization structure” is a bit misleading, the author shall further clarify more details.

9.      Several typos or vagueness shall be further clarified:

a. Line 120, ND are sp2-type nanocarbons

b. Line 493, a lattice spacing of 002

Author Response

Reviewer 3

Comments and Suggestions for Authors

The manuscript deals with the Carbon-based Nanomaterials for Catalytic Wastewater Treatment: A Review. The manuscript is well structured; however, some points should be clear for better understanding of the presented data for possible publication in Molecules.

  1. In introduction, the author should clearly mention the utilization of Carbon-based materials and their importance. The author should carefully read the articles like Advanced Powder Technology 32 (2021) 3770–3787; Environmental Research 215 (2022) 114140 and refine this portion accordingly.

Answer: We appreciate the reviewer’s comments regarding the explanation about the utilization of Carbon-based materials and their importance in the introduction part of the revised manuscript and new references are added in reference [17-21].

  1. In the introduction section, the author should include dyes, their role, and their hazardous effect on the ecosystem through carefully reading the article Advanced Powder Technology 33 (2022) 103708, https://doi.org/10.1007/s11356-022-22144-3.

Answer: Thank you so much for the suggestions. We have inserted the hazardous effect on the ecosystem through and added new references 11and 12.

  1. The author should clearly explain the enhanced efficiency of the current materials under study.

Answer: Thank you so much. We improved our explanation of the current studies materials in every section of the revised review manuscript.

  1. All figures should be in the same size.

Answer: Thank you so much. We corrected the figure size.

  1. It is very difficult and expensive to synthesize carbon-based material; what is the advantage of using these materials?

 Answer: Thank you so much. In the introduction part and conclusion part we have discussed about advantages of the carbon-based materials.

  1. What is the novelty of the present findings?

Answer: Thank you so much. We explained the novelty and findings of revised review manuscript in the introduction section and conclusion section.

Round 2

Reviewer 1 Report

I am quite happy with the revised manuscript and recommend it for publication.